# A Preclinical Model for Parkinson’s Disease Based on Transcriptional Gene Activation via KEAP1/NRF2 to Develop New Antioxidant Therapies

**DOI:** 10.3390/antiox12030673

**Published:** 2023-03-09

**Authors:** Juan Segura-Aguilar, Bengt Mannervik

**Affiliations:** 1Molecular & Clinical Pharmacology, ICBM, Faculty of Medicine, University of Chile, Independencia, Santiago 8380000, Chile; 2Department of Biochemistry and Biophysics, Arrhenius Laboratories, Stockholm University, SE-10691 Stockholm, Sweden; 3Department of Chemistry, Scripps Research, La Jolla, CA 92037, USA

**Keywords:** dopamine, Parkinson’s disease, neuromelanin, antioxidants, aminochrome, glutathione transferase M2-2, DT-diaphorase, ferroptosis, KEAP1/NRF2, dopaminergic neurons

## Abstract

Investigations of the effect of antioxidants on idiopathic Parkinson’s disease have been unsuccessful because the preclinical models used to propose these clinical studies do not accurately represent the neurodegenerative process of the disease. Treatment with certain exogenous neurotoxins induces massive and extremely rapid degeneration; for example, MPTP causes severe Parkinsonism in just three days, while the degenerative process of idiopathic Parkinson´s disease proceeds over many years. The endogenous neurotoxin aminochrome seems to be a good alternative target since it is formed in the nigrostriatal system neurons where the degenerative process occurs. Aminochrome induces all the mechanisms reported to be involved in the degenerative processes of idiopathic Parkinson’s disease. The presence of neuromelanin-containing dopaminergic neurons in the postmortem brain of healthy elderly people suggests that neuromelanin synthesis is a normal and harmless process despite the fact that it requires oxidation of dopamine to three ortho-quinones that are potentially toxic, especially aminochrome. The apparent contradiction that neuromelanin synthesis is harmless, despite its formation via neurotoxic ortho-quinones, can be explained by the protective roles of DT-diaphorase and glutathione transferase GSTM2-2 as well as the neuroprotective role of astrocytes secreting exosomes loaded with GSTM2-2. Increasing the expression of DT-diaphorase and GSTM2-2 may be a therapeutic goal to prevent the degeneration of new neuromelanin-containing dopaminergic neurons. Several phytochemicals that induce DT-diaphorase have been discovered and, therefore, an interesting question is whether these phytochemical KEAP1/NRF2 activators can inhibit or decrease aminochrome-induced neurotoxicity.

## 1. Mechanisms Involved in Neurodegeneration in Parkinson’s Disease

Although 65 years have passed since the discovery that the motor symptoms of idiopathic Parkinson’s disease are related to the massive loss of neuromelanin-containing neurons in the nigrostriatal system, it is still unclear what triggers the degenerative process of this neuronal system [1,2]. The first widely studied mechanism was the role of oxidative stress in this disease, then mitochondrial dysfunction, and, in the early 1990s, the existence of a mutation in the alpha-synuclein gene was discovered in some families with genetic Parkinson’s. This discovery was a major boost in basic research as it was the first gene with mutations that induced familial or genetic Parkinson’s [3]. Subsequently, more genes related to familial Parkinson’s disease were identified, such as the parkin gene, PINK-1, LRRK2, VPS35, DJ1, ATP13A2, etc. [4]. Although the origin of neurodegeneration in the nigrostriatal system is unclear, there is a general consensus that certain mechanisms are involved in the loss of neuromelanin-containing dopaminergic neurons. They are related to the appearance of motor symptoms and involve oxidative stress, mitochondrial dysfunction, aggregation of alpha-synuclein to neurotoxic oligomers, dysfunction of both lysosomal and proteasomal protein degradation systems, endoplasmic reticulum stress, and neuroinflammation [5,6,7,8].

## 2. Clinical Studies to Develop Antioxidant Therapies

The recognition of oxidative stress in the degenerative process of the nigrostriatal system has been a focus for several decades. The discovery that the motor movements of Parkinson’s disease are related to the loss of the neuromelanin-containing dopaminergic neuron of the nigrostriatal system that generates a significant drop in the release of dopamine has implied that preclinical models based on exogenous neurotoxins such as 6-hydroxydopamine have played an important role in the investigation of this degenerative process and the development of potential therapies. There are two sources of oxidative stress generation in dopaminergic neurons, one being the oxidative deamination of dopamine catalyzed by the enzyme monoamine oxidase that generates ammonia and hydrogen peroxide [9], which in the presence of reduced iron generates hydroxyl radicals and therefore profuse oxidative stress within dopaminergic neurons. Another specific source of oxidative stress within dopaminergic neurons arises during the synthesis of neuromelanin which requires the oxidation of dopamine to three ortho-quinones (dopamine ortho-quinone, aminochrome, and 5,6-indolequinone). Aminochrome reduction with one electron generates leukoaminochrome *o*-semiquinone radical that is extremely reactive with oxygen [10] generating oxidative stress in neuromelanin-containing dopaminergic neurons of the nigrostriatal system [11]. Another source of permanent oxidative stress, although not specific for dopaminergic neurons, is the leakage of electrons from the mitochondrial electron transport chain during the generating of ATP through the oxidative phosphorylation of ADP [12]. 

Toxicity due to oxidative stress has prompted major efforts to developing antioxidant therapies using successful results in preclinical models with exogenous neurotoxins. Coenzyme Q10 works as a scavenger of reactive species during oxidative stress [13] and has also been shown to reduce the degenerative effects of MPTP in mice, supporting the idea of conducting clinical studies in patients with Parkinson’s [14]. Coenzyme Q10 also protects against 6-hydroxydopamine neurotoxicity [15]. Preclinical studies and a clinical study phase with coenzyme Q suggested clinical benefits for Parkinson’s patients. However, a randomized phase III clinical study was performed with 600 participants administered placebo, 1200 mg/d of CoQ10, or 2400 mg/d of CoQ10, and in addition all participants received 1200 IU/d of vitamin E. This study concluded that this treatment has no benefits for patients with Parkinson’s disease [16]. Another study in 2015, involving reduced coenzyme Q10, was carried out with 10 Parkinson’s patients and showed positive effects [17]. However, we must take into account the chemical characteristics of reduced coenzyme Q10 that after being administered to the patient will be oxidized again inside the neuron. A coenzyme Q10 analogue (MitoQ10) obtained by binding coenzyme Q10 with triphenylphosphonium cation has been shown to have protective effects in preclinical models of MPTP [18] and 6-hydroxydopamine [19,20]. However, a double-blind study with 120 patients with newly diagnosed Parkinson’s disease without treatment did not demonstrate clinical benefits [21].

Urate is another antioxidant that has been used in clinical studies with Parkinson’s patients. Urate is a product of purine metabolism and, in preclinical studies with 1-methyl-4-phenyl-1,2,5,6-tetrahydropyridine (MPTP)-induced oxidative stress, urate showed an antioxidant effect. Urate is also neuroprotective in a 6-hydroxydopamine-lesioned rat model linked to the Akt/GSK3β signaling pathway [22]. Protective effects were also shown in preclinical models involving MPTP; urate increases the expression of γ-glutamate-cysteine ligase catalytic subunit, heme oxygenase-1, and DT-diaphorase via NFR2 [23,24]. However, a phase 3 study with 298 pharmacologically untouched Parkinson’s patients treated with the urate precursor inosine for up to 2 years to increase urate levels showed no clinical benefit [25].

## 3. Neurotoxins in Preclinical Models for Parkinson’s Disease

The lessons that we can draw from these preclinical models, based on exogenous neurotoxins causing a massive and extremely rapid degeneration, are that: (i) A preclinical model must resemble the extremely slow degenerative process in idiopathic Parkinson’s has to be different. The triggering of all the mechanisms that have been observed to be involved in this degenerative process occurs at different time scales. For this reason, the neurotoxin for a preclinical model to study the mechanisms and be able to test new drugs for idiopathic Parkinson’s must have other properties, presumably of endogenous origin. (ii) The neuronal death induced by this endogenous neurotoxin cannot be expansive or massive but must be focused on individual dopaminergic neurons containing neuromelanin. Only such a model would mimic the extremely slow degenerative process and progression of the disease that takes many years.

### 3.1. Exogenous Neurotoxins in Preclinical Models for Parkinson’s Disease

Most of the preclinical studies that have contributed to new clinical trials in the search for new pharmacological treatments for modifying or halting the progress of Parkinson’s disease have used 6-hydroxydopamine and subsequently MPTP. 6-hydroxydopamine has high affinity to transporters of dopamine and noradrenaline, inducing neurotoxicity in both dopaminergic and noradrenergic neurons, and its use in animals for Parkinson’s experimental models requires the presence of an inhibitor of the noradrenaline transporter to obtain specificity for dopaminergic neurons [26,27]. A major characteristic of 6-hydroxydopamine is that it is unstable in the presence of oxygen, which implies that its injection into the striatum, substantia nigra, or the axonal bundle requires the use of ascorbic acid to prevent its autoxidation before being transported to dopaminergic neurons. 6-hydroxydopamine in the cytosol of dopaminergic neurons is autoxidized, generating a semiquinone radical that is also extremely reactive with oxygen, generating oxidative stress [28,29]. This relationship of oxidative stress with the massive degeneration of neuromelanin-containing dopaminergic neurons of the nigrostriatal system motivated the development of clinical studies with antioxidants. 6-hydroxydopamine is the source of oxidative stress in this preclinical model, but this molecule does not exist in the human brain. The difference between 6-hydroxydopamine and MPTP is that the latter has been shown to generate Parkinsonism in humans when drug addicts have used illegal drugs contaminated with MPTP. However, these two exogenous neurotoxins have in common that they cause extremely rapid and massive degeneration of the nigrostriatal system. MPTP induced severe Parkinsonism to drug addicts in just 3 days [30], which is in contrast to the extremely slow progression of idiopathic Parkinson’s disease. The latter takes years to develop motor symptoms, which emerge when 60% of the neuromelanin-containing neurons of the nigrostriatal system have disappeared. After the onset of motor symptoms, at least 15 years may pass before death. The question is whether investigations with exogenous neurotoxins can serve as a preclinical model to study the mechanisms of the disease and also test possible drugs for a treatment that can modify the course of the disease. The number of dopaminergic neurons of the nigrostriatal system has been estimated to be between 800,000 to 1,000,000, counting both sides of the brain [31]. If we consider that the motor symptoms appear when 60% of the neuromelanin-containing dopaminergic neurons have degenerated, these patients have between 360,000 to 400,000 surviving neurons. This implies that if a patient survives for around 15 years, between 58 and 73 dopaminergic neurons that contain neuromelanin die each day. By contrast, a subchronic animal model with one daily injection of MPTP for 5 days induces a 50% loss of dopaminergic neurons in only 28 days [32]. It is difficult to apply the results of such an extremely rapid cell death as a preclinical model compared with the extremely slow progression of nigrostriatal degeneration in idiopathic Parkinson’s disease. In a clinical study that, for example, uses coenzyme Q10, if it inhibited 100% oxidative stress and the death of dopaminergic neurons, this antioxidant would prevent the death of 0.0073% of these neurons per day or 1.3% in 6 months. The question is, can this result be significant for delaying or halting the progression of idiopathic Parkinson’s disease? It is doubtful whether successful results would be obtained in clinical studies using exogenous neurotoxins that induce an extremely rapid and massive degenerative process that is the complete opposite of what occurs in the development of idiopathic Parkinson’s disease.

### 3.2. Endogenous Neurotoxins as Preclinical Model for Parkinson’s Disease

The question is: what endogenous neurotoxins, formed in neuromelanin-containing dopaminergic neurons, might serve in preclinical models for idiopathic Parkinson’s disease? There are three endogenous neurotoxins that can be involved in the degeneration of dopaminergic neurons containing neuromelanin:

#### 3.2.1. Neurotoxic Oligomers of Alpha-Synuclein

Alpha-synuclein is a protein that exists in a monomeric form and, under certain conditions, can aggregate to form fibrils, which are deposited in Lewy bodies seen in postmortem tissue [33]. However, the formation of these neurotoxic oligomers requires the existence of a mutation in familial Parkinson’s disease and, in the case of idiopathic Parkinson’s, requires the existence of products of the oxidation of dopamine to neuromelanin, such as aminochrome [34]. Another argument against alpha-synuclein as the neurotoxin that triggers all the degenerative mechanisms reported in idiopathic Parkinson’s is that the expression of alpha-synuclein is not limited to the nigrostriatal system, since alpha-synuclein is also expressed in other regions such as the cerebellum and the cortex [35]. It has been postulated that the aggregation of alpha-synucleins to fibrils and their accumulation in Lewy bodies could be responsible for the progression of the disease from one region to another region of the brain. The neurons that form these Lewy bodies secrete them through exosomes that penetrate neighboring neurons. This dissemination of Lewy bodies containing alpha-synuclein fibrils would allow spreading from one region to another region [33]. This hypothesis is controversial, because, if Lewy bodies were to participate in the spread of the disease, why are these observed in postmortem tissues? When a neuron dies, it is phagocytosed by the microglia, eliminating all traces of Lewy bodies formed through years of degeneration [36]. Therefore, the postmortem tissue represents the neurons surviving the degenerative process that has existed for years. This rather suggests that the formation of Lewy bodies is a neuroprotective mechanism to prevent alpha-synuclein from aggregating to neurotoxic oligomers [37]. An additional argument against the hypothesis of the spread of idiopathic Parkinson’s disease through Lewy bodies migrating from one neuron to neighboring neurons and subsequently from one region to another region of the brain is the propagative process that not only affects a single neighboring neuron but hundreds of neurons existing around the neuron that secretes exosomes containing Lewy bodies. 

#### 3.2.2. 3,4-Dihydroxyphenylacetaldehyde

3,4-Dihydroxyphenylacetaldehyde also known as DOPAL, is a product of the oxidative deamination of dopamine catalyzed by the enzyme monoamine oxidase that generates hydrogen peroxide and ammonium [38]. The enzyme aldehyde dehydrogenase-1 participates in the further degradation of dopamine, which converts DOPAL into 3,4-dihydroxyphenylacetic acid. In a study of protein expression in postmortem material, it was observed that the levels of aldehyde dehydrogenase-1 were lower in Parkinson’s patients than in controls [39]. DOPAL accumulation could be neurotoxic because DOPAL induces oxidative stress and alpha-synuclein aggregation [38]. The problem is that the postmortem tissue is the tissue surviving the degenerative process during many years of Parkinson’s disease and, if there is a low expression of aldehyde dehydrogenase-1 in these tissues surviving years of a degenerative process, DOPAL cannot play a neurotoxic role in dopaminergic neurons that contain neuromelanin.

#### 3.2.3. Aminochrome

Aminochrome is an endogenous neurotoxin formed in the synthesis of neuromelanin in dopaminergic neurons that contain neuromelanin, which are precisely the neurons that disappear in Parkinson’s disease. To synthesize neuromelanin, dopamine needs to be oxidized sequentially to three transient ortho-quinones (dopamine ortho-quinone, aminochrome, and 5,6-indolequinone), among which the latter is finally polymerized to form the pigment neuromelanin. Neuromelanin is a dark pigment contained in membrane vesicles called neuromelanin-containing organelles with a diameter of 200–600 nm that, besides neuromelanin, contain lipids and metals such as S, Fe, and Cu in healthy subjects [40]. A study carried out with infrared spectroscopy in different regions of the human brain revealed that the neuromelanin of the regions that degenerate in Parkinson’s disease, substantia nigra and locus coeruleus, have a similar structure but different from those regions not affected by the disease [41]. The synthesis of neuromelanin is a normal and harmless process that also has a neuroprotective role by preventing the neurotoxic effects of reactive metals. However, neuromelanin released from dead dopaminergic neurons is able to activate microglia, generating proinflammatory molecules and reactive oxygen and nitrogen species [42]. Neuromelanin injected into the substantia nigra of rats induces microglia activation and loss of tyrosine hydroxylase-positive neurons [43]. All three ortho-quinones derived from dopamine are neurotoxic, but the most neurotoxic and most long-lived is aminochrome. Aminochrome induces oxidative stress, neuroinflammation, mitochondrial dysfunction, alpha-synuclein aggregation to neurotoxic oligomers, dysfunction of both lysosomal and proteasomal protein degradation systems, and plasma reticulum stress [11,34,44,45,46,47,48,49,50,51]. Due to the chemical characteristics of aminochrome, it does not induce expansive neurotoxic effects such as alpha-synuclein fibrils. Aminochrome has been shown to be stable for up to 40 min before its conversion to 5,6-indolequinone begins [52]. 

## 4. Preclinical Model for Idiopathic Parkinson’s Disease

The preclinical model for idiopathic Parkinson’s disease should use (i) an endogenous neurotoxin that is formed within the neuromelanin-containing dopaminergic neurons of the nigrostriatal system; (ii) a neurotoxin that induces a focal (non-expansive) degeneration; (iii) it should trigger all the mechanisms involved in the degenerative process, such as oxidative stress, mitochondrial dysfunction, alpha-synuclein aggregation to neurotoxic oligomers, protein degradation dysfunction (lysosomal and proteasomal system), endoplasmic reticulum stress, and neuroinflammation. As we have seen previously, the only one of the endogenous neurotoxins that we have previously described that meets these requirements is aminochrome. 

Aminochrome is neurotoxic by being one-electron reduced to leukoaminochrome *o*-semiquinone radical by most flavoenzymes with the exception of DT-diaphorase (NADP(H):quinone oxidoreductase), which is the unique flavoenzyme that reduces quinones with two electrons [53]. In studies performed with electron spin resonance (ESR), NADPH cytochrome P450 reductase was found to reduce both dopamine ortho-quinone and aminochrome to the free radicals dopamine *o*-semiquinone and leukoaminochrome *o*-semiquinone, respectively. The ESR spectrum of dopamine *o*-semiquinone and of leukoaminochrome *o*-semiquinone radical was detected at 1 min. However, at 2 min only the dopamine *o*-semiquinone radical spectrum was detectable by ESR due to the high reactivity of leukoaminochrome *o*-semiquinone radical with oxygen [10]. Aminochrome one-electron reduction to the leukoaminochrome *o*-semiquinone radical induces oxidative stress due to the extremely rapid reaction with oxygen, which causes re-oxidation to aminochrome. This generates a redox cycling between aminochrome and leukoaminochrome *o*-semiquinone radical that functions until cellular dioxygen and/or NAD(P)H are depleted. This redox cycling generates oxidative stress and the depletion of cytosolic NADH for its use to generate ATP in the mitochondrial electron transport chain (Figure 1). 

Aminochrome is also neurotoxic by forming covalent complexes with proteins such as alpha-synuclein that induce the formation of neurotoxic oligomers [34]. Aminochrome forms adducts also with actin and α- and β-tubulin disrupting the cytoskeleton architecture [54], which plays an important role in microtubule formation [55]. Aminochrome inhibits complex I of the mitochondrial respiratory chain, resulting in the inhibition of ATP production and mitochondrial dysfunction [45,46]. Aminochrome induces protein degradation dysfunction by inhibiting lysosomal and proteasomal systems [48,49,50]. Aminochrome induces lysosomal dysfunction by inhibiting the vacuolar-type H^+^-ATPase that plays an essential role in maintaining an acidic pH by pumping protons into lysosomes [56]. Aminochrome also induces endoplasmic reticulum stress and neuroinflammation [44,47,48]. 

Aminochrome in vivo induces neuronal dysfunction as a consequence of a significant decrease in dopamine release with concomitant enhanced GABA release [57]. Aminochrome induces mitochondrial dysfunction with a significant decrease in ATP production, affecting axonal transport of monoaminergic vesicles for neurotransmission to the terminals that explain the significant decrease in the number of these vesicles in the terminals compared with saline as control. Aminochrome induces a progressive degeneration of dopaminergic neurons while the striatal dopaminergic fibers are intact, accompanied by a dramatic change of tyrosine positive neuron morphology, a phenomenon known as cell shrinkage [57]. 

The ideal model for idiopathic Parkinson’s disease should consider that the neurotoxin that triggers the degenerative process is generated within the same neuromelanin-containing dopaminergic neuron and induces a degenerative process focused on a single neuron. This last requirement is practically impossible to achieve, since the injection of aminochrome into the striatum or substantia nigra will affect all the neurons that are in contact with aminochrome solution. However, if aminochrome is the neurotoxin that triggers neurodegeneration in idiopathic Parkinson’s, this model could be a good target to search for possible pharmacological compounds that prevent, inhibit, or slow down this degenerative process in idiopathic Parkinson’s. 

There is a study in which the human enzyme tyrosinase was overexpressed in the substantia nigra of rats, exacerbating the production of neuromelanin in this region of the brain in all types of neurons and glial cells, inducing nigrostriatal neurodegeneration, hypokinesia, and Lewy body-like formation [58]. The problem with this model is that (i) this overexpression affects the majority of dopaminergic neurons, generating a massive effect on the degenerative process; (ii) the oxidative effect of tyrosinase is not specific for dopaminergic neurons since this overexpression affects all neurons and glial cells. Tyrosinase catalyzes the oxidation of monophenols or diphenols in the presence of dioxygen without the need for a cofactor; however, its catalytic activity with diphenols is higher than with monophenols [59]. Therefore, indiscriminate overexpression in all types of neurons and glial cells reduces the specificity of the model, since tyrosinase can oxidize monophenols in other types of neurons or glial cells with unknown effects. In the case of overexpression of tyrosinase in astrocytes, this enzyme can also oxidize dopamine, since this type of glial cell can take up dopamine released during neurotransmission; (iii) the triggering agent of the neurodegenerative process in this preclinical model is not a neurotoxin that can be the target of new pharmacological drugs attempting to stop the degenerative process, but rather it is tyrosinase, which is not expressed in the human substantia nigra. However, there are some reports that suggest its expression in this tissue [60], but if it were true that tyrosinase is normally expressed in neuromelanin-containing dopaminergic neurons, the presence of the pigment would be observed from an early age in children. Several studies have indeed shown that tyrosinase is not present in the human substantia nigra, even using highly specific and sensitive mass spectroscopy [61,62] (Zucca et al., 2018; Tribl et al., 2007). It has been proposed that animals that produce more neuromelanin and more closely resemble the human brain such as sheep and goats are more appropriate as a preclinical animal model for Parkinson’s disease [63] (Capucciati et al., 2021).

## 5. Transcriptional Gene Activation via KEAP1/NRF2

Transcriptional activation of genes encoding a battery of cellular defense enzymes that provide protection against oxidative and electrophilic stress is affected by the transcription factor NRF2 (nuclear factor (erythroid-derived 2)-like 2) binding to an antioxidant responsive element or an electrophile responsive element (ARE/EpRE). Among the various cytoprotective proteins are γ-glutamylcysteine ligase, GSTs, and DT-diaphorase [18,64]. NFR2 is bound to KEAP1 (Kelch-like ECH-associated protein 1) in the cytosol, where it is directed to rapid proteasomal degradation. KEAP1 senses reactions of its sulfhydryl groups with electrophiles and oxidants and, as a consequence, NRF2 is released and targets ARE/ErRE elements of DNA in the nucleus. KEAP1/NRF2 is recognized as the most prominent protective system against electrophilic and oxidative insults; the upregulation of key enzymes in the defense against toxicants implicated in Parkinson’s disease and other neurodegenerative conditions is noteworthy. For obvious reasons, the KEAP1/NFR2 system has been considered for pharmacological interventions in Parkinson’s disease [19,65], Figure 2.

## 6. Neuroprotection against Neurodegeneration of Nigrostriatal System in Parkinson’s Disease

Studies carried out with postmortem material from healthy elderly people show intact neuromelanin-containing dopaminergic neurons, suggesting that neuromelanin synthesis is a natural and harmless process. Neuromelanin has been observed to increase over the years in the human brain [66]. However, there is an apparent contradiction, since the synthesis of neuromelanin requires the oxidation of dopamine to three ortho-quinones that can be neurotoxic. Among these aminochrome is the most toxic that induces oxidative stress, the formation of neurotoxic alpha-synuclein oligomers, dysfunction of both lysosomal and proteasomal protein breakdown systems, plasma reticulum stress, mitochondrial dysfunction, and neuroinflammation. The reason why the elderly have their neuromelanin-containing dopaminergic neurons intact at the time of death is because there are two enzymes that prevent the neurotoxic effects of aminochrome DT-diaphorase and glutathione transferase M2-2.

### 6.1. DT-Diaphorase

DT-diaphorase (NAD(P)H: quinone oxidoreductase, NQO1) is a flavoenzyme with FAD as a prosthetic group that transfers two electrons from either NADH or NADPH to quinones, thereby reducing them to hydroquinones [53,61,67]. This enzyme is 90% located in the cytosol and 5% associated with the endoplasmic reticulum and mitochondria. It is widely expressed in different organs; in the brain it is expressed in the substantia nigra, striatum, cortex, hypothalamus, and hippocampus. In rat substantia nigra, DT-diaphorase is responsible for 97% of the total quinone reductase activity. DT-diaphorase is expressed in tyrosine hydroxylase positive neurons and in astrocytes. DT-diaphorase reduces aminochrome to leukoaminochrome, preventing its neurotoxic effects in a catecholaminergic cell line [68], but DT-diaphorase also prevents aminochrome induced-toxicity in a human astrocyte cell line [69]. DT-diaphorase prevents aminochrome-induced oxidative stress, mitochondrial dysfunction, formation of neurotoxic alpha-synuclein oligomers, proteasome dysfunction, autophagy dysfunction, and lysosome dysfunction [11,26,34,40,47,48,49,51,54,56,62,64].

### 6.2. Glutathione Transferase M2-2

Electrophilic compounds such as epoxides, alkenals, and quinones can react with GSH and thereby become inactivated and form glutathione conjugates suitable for export from the cytosol [12,70]. The reactions are catalyzed by 20-odd enzymes, which are differentially distributed in cells and tissues. In humans and other mammals, the GSTs have been grouped into membrane-bound (microsomal) and soluble (cytosolic) proteins [13,71]. The former are members of the MAPEG (membrane-associated proteins in eicosanoid and glutathione metabolism) family [14,72]. The soluble GSTs are dimeric proteins occurring in eight classes of homologous sequences, which are catalytically active as homodimers as well as heterodimers [15,73]. The dimers are thus denoted according to their subunit composition, for example, as GST M1-1, GST M1-2, and GST M2-2 encoded by the GSTM1 and GSTM2 genes of the Mu class. With respect to the large variety of compounds identified as substrates for the various GSTs, it is noteworthy that only GST M2-2 has been found to catalyze with high efficiency the conjugation of ortho-quinones derived from dopamine [17,74,75]. The most active glutathione transferase catalyzing aminochrome conjugation is glutathione transferase M2-2, which gives rise to 4-S-glutathionyl-5,6-dihydroxyindoline. Interestingly, 4-S-glutathionyl-5,6-dihydroxyindoline is not oxidized by physiologically occurring oxidants such as dioxygen, hydrogen peroxide, and superoxide. Glutathione transferase M2-2 also catalyzes glutathione conjugation of the aminochrome precursor dopamine ortho-quinone to 5-glutathionyl dopamine that is degraded to 5-cysteinyl dopamine, which has been detected in human cerebrospinal fluid and neuromelanin [16,17,72,73,74,75,76,77,78] (Figure 3). 

However, whether 5-cysteinyl is an end product is controversial as it has been reported by in vitro experiments that 5-cysteinyldopamine may be neurotoxic by being oxidized to 5-cysteinyldopamine o-quinone, which is converted to a bicyclic o-quinone imine [76,79]. If this mechanism is correct, it remains to explain the presence of 5-cysteinyldopamine in human cerebrospinal fluid and neuromelanin [77,78], since 5-cysteinyldopamine is converted to bicyclic o-quinone imine [79].

### 6.3. Astrocytes Protect Dopaminergic Neurons against Aminochrome Neurotoxicity

The brain is responsible for more than 20% of the energy consumption in the human body. The organism derives ATP via oxidation of carbohydrates, fats, and proteins and by coupling electron transport with phosphorylation of ADP to ATP. Neurons require ATP to transport proteins and neurotransmission vesicles to neuronal terminals, and neurotransmission itself is completely dependent on ATP. In other words, neuronal function is absolutely dependent on the presence of ATP, which, to a large extent, is generated in the mitochondria. Electrons in the transport chain from NADH flowing through complex I or complex II are further transmitted to ubiquinone Q10 (also known as coenzyme Q10), which in turn transfers the electrons to complex III. In this transfer of electrons, ubiquinone Q10 is reduced with one electron to the free radical ubiquinone Q10 semiquinone. The latter is subsequently reduced to ubiquinone Q10 hydroquinone, which transfers the electrons to complex III. However, the radical ubiquinone Q10 semiquinone is able to reduce dioxygen to superoxide, generating electron transport leakage and oxidative stress. Superoxide enzymatically or spontaneously can generate hydrogen peroxide, which, in the presence of reduced iron, forms hydroxyl radicals. This leakage of electrons from the mitochondrial electron transport chain permanently exposes neurons to conditions of oxidative stress. Astrocytes play an important role in protecting neurons against oxidative stress since astrocytes secrete precursors of glutathione, which is an important antioxidant [77,80,81]. Neuromelanin-containing dopaminergic neurons are also exposed to the neurotoxic effects of aminochrome, leading to oxidative stress, mitochondrial dysfunction, formation of neurotoxic alpha-synuclein oligomers, dysfunction of both lysosomal and proteasomal protein degradation systems, neuroinflammation, and endoplasmic reticulum stress [44,45,46,47,48,49,62,63]. However, astrocytes also play a neuroprotective role for neuromelanin-containing dopaminergic neurons by secreting exosomes loaded with the enzyme glutathione transferase M2-2 that penetrate dopaminergic neurons to, in concert with DT-diaphorase, prevent the neurotoxic effects of aminochrome [73,77,78,80,81,82] (Figure 4). 

### 6.4. Glutathione and Oxidative Stress

Glutathione (GSH) is a fundamental endogenous constituent of the molecular defense that provides protection against electrophilic agents and oxidative stress [1,83]. GSH, L-γ-glutamyl-L-cysteinylglycine, is formed by the sequential actions of γ-glutamylcysteine ligase and glutathione synthetase, of which reactions the first is rate-limiting in GSH biosynthesis [2,84]. It has been demonstrated that GSH is ubiquitously present in the brain and that substantia nigra in Parkinson’s patients is selectively depleted in GSH content compared with healthy individuals [3,4,85,86]. Whether the loss of GSH is a cause or an effect of the disease remains unknown but attempts to raise the GSH concentration by pharmacological intervention have been made under the premise that an elevated GSH level would provide a therapeutic consequence [5,87]. In general, GSH cannot cross cell membranes or penetrate the blood–brain barrier and oral administration has been ruled out. Nasal spray delivery of GSH has been attempted, but the efficacy is unclear [6,88]. The amino acid limiting the biosynthesis of GSH is cysteine, which is normally transported into cells as the disulfide cystine via the system xc-cystine/glutamate antiporter [7,89]. Thus, intravenous administration of the precursor N-acetylcysteine has been reported to increase the GSH concentration in the brain [8,90]. The characteristic pathological features of substantia nigra in Parkinson’s patients comprise enhanced iron deposition and lipid peroxidation, as wells as defects in the transport system xc-. These attributes are congruent with the hallmarks of ferroptosis, which is currently receiving attention in various diseases [9,91]. Preventing the progressive loss of dopaminergic neurons by interfering with ferroptosis is a new approach to treatment [6,10,88,92]. GSH can serve as an antioxidant and five selenium-dependent peroxidases (GPXs) catalyze the reduction of hydrogen peroxide and various lipid hydroperoxides [11,93]. The enzymes play a prominent role in the protection against reactive oxygen species (ROS). GPX4 is particularly active with phospholipid hydroperoxides and appears to play an essential function in preventing ferroptosis [9,91].

## 7. Conclusions and Future Directions

It is clear that oxidative stress plays a role in the degenerative process of the nigrostriatal system in idiopathic Parkinson’s disease. In our opinion, the failure of clinical studies with antioxidants is due to the fact that the preclinical models used to propose these clinical studies are based on exogenous neurotoxins (MPTP and 6-hydroxydopamine) that do not represent what happens in the neurodegenerative process of idiopathic Parkinson’s disease. These neurotoxins induce massive and extremely rapid degeneration; for example, MPTP induces severe Parkinsonism in just three days, while the degenerative process takes many years. How to test a possible new drug for idiopathic Parkinson’s disease if the preclinical model does not include what triggers the degenerative process? The drug examined for potential use in the disease cannot inhibit the degenerative process, since these exogenous neurotoxins do not exist in the human nigrostriatal system. Aminochrome seems to be a good alternative, since it is formed in the nigrostriatal neurons where the degenerative process occurs and induces all the mechanisms that have been reported to be involved in the degenerative process of idiopathic Parkinson’s disease, such as oxidative stress, dysfunction mitochondrial, formation of neurotoxic alpha-synuclein oligomers, endoplasmic reticulum stress, neuroinflammation, and dysfunction of both lysosomal and proteasomal protein degradation systems. Unfortunately, aminochrome must be injected into the striatum or substantia nigra and a neuron-in-neuron focal degeneration such as occurs in the disease cannot be reproduced. However, although the injection of aminochrome affects many neurons immediately, it produces a progressive neuronal dysfunction that progresses slowly. In addition, if aminochrome is what triggers the degeneration of neuromelanin-containing dopaminergic neurons in idiopathic Parkinson’s disease, this preclinical model could serve to detect potential new drugs that change the progress of the disease or stop the neurodegenerative process.

The presence of neuromelanin-containing dopaminergic neurons in the postmortem brain of healthy elderly people suggests that neuromelanin synthesis is a normal and harmless process, despite the fact that it requires the oxidation of dopamine to three ortho-quinones that are potentially toxic, especially aminochrome. This apparent contradiction that neuromelanin synthesis is harmless despite the formation of neurotoxic ortho-quinones can be explained by the neuroprotective role of DT-diaphorase and GSTM2-2 and the neuroprotective role of astrocytes secreting exosomes loaded with GSTM2-2. However, the neuroprotective capacity of DT-diaphorase and GSTM2-2 is not infinite but depends on their Km with respect to aminochrome. Excess aminochrome formation that exceeds the neuroprotective capacity of these enzymes may explain the degenerative process in idiopathic Parkinson’s disease. Aminochrome is neurotoxic by inducing the formation of neurotoxic alpha-synuclein oligomers, mitochondrial dysfunction, oxidative stress, dysfunction of both lysosomal and proteasomal protein degradation systems, endoplasmic reticulum stress, and neuroinflammation. Therefore, increasing the expression of DT-diaphorase and GSTM2-2 may be a therapeutic modality to prevent the degeneration of new neuromelanin-containing dopaminergic neurons. The activation of NFR2 induces the expression phase 2 enzymes, including both DT-diaphorase and GST [79,92], and therefore its activation may be the target of potential new drugs that slow down the neurogenerative process in idiopathic Parkinson’s disease. Several phytochemicals in natural products that have different biological actions have been shown to be activators of NFR2. Sesamol is a component of sesame oil having an anti-inflammatory and antioxidant effect that has been proposed as a cardioprotector [80,93]. A component of Rhododendron l. is farrerol that has been shown to have an anti-inflammatory, antioxidant, and neuroprotective effect [81,94]. Oxyphylla A is a component of Alpinia oxyphylla that is used in traditional Chinese medicine to treat memory loss and has been shown to decrease cognitive deficits and improve muscle strength [82,95]. Sargahydorquinoic acid, isolated from the marine alga Sargassum serratifolium, has been shown to be a potent antioxidant [83,96]. Ginsenoside Re has been isolated from the stem, berry, leaf, flower bud, and root of Panax ginseng and has exhibited a long list of pharmacological effects including neuroprotective effects [84,85,97,98]. Sulforaphane, found in cruciferous vegetables such as broccoli, Brussels sprouts, cabbage, and cauliflower, has anti-apoptotic, antioxidant, and anti-inflammatory properties [86,99]. Protopanaxatriol, present in Panax ginseng Mayer, protects against oxidative stress [87,100]. The flavonoids naringin and naringenin, present in tomatoes, bergamots, and various citrus, have anti-inflammatory, antioxidant, and neuroprotective properties [88,101,102,103,104,105]. Several phytochemicals inducing DT-diaphorase have been reported that include farrerol, sesamol, oxyphylla A, sargassum serratifolium, ginsenoside Re, sulforaphane, protopanaxatriol, and naringin [80,81,82,83,84,85,86,87,88,89,90,91,92,93,94,95,99]. Therefore, an interesting question is whether these phytochemical NFR2 activators can inhibit or decrease aminochrome-induced neurotoxicity.

## Figures and Tables

**Figure 1 antioxidants-12-00673-f001:**
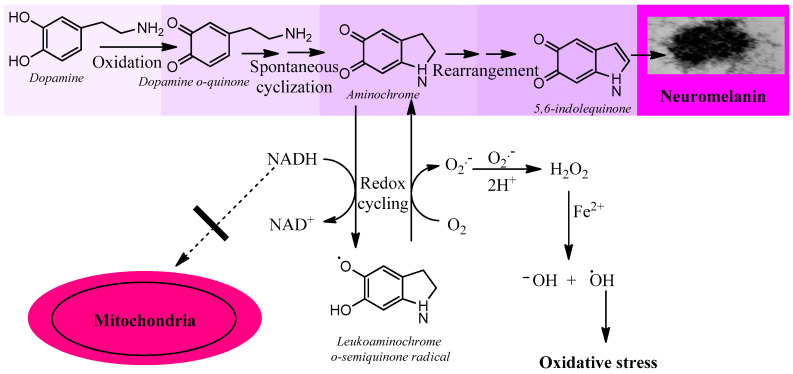
Aminochrome induces oxidative stress. Aminochrome can be reduced with one electron to produce the leukoaminochrome *o*-semiquinone radical by flavoenzymes using the cytosolic NADH generated to be used in the mitochondria to generate ATP through the transport chain coupled to oxidative phosphorylation from ADP to ATP. The leukoaminochrome *o*-semiquinone radical is extremely unstable in the presence of dioxygen and autoxidizes to regenerate aminochrome, which is reduced again, creating a redox cycle between aminochrome and leukoaminochrome *o*-semiquinone radical. This redox cycling is very fast and depletes as much oxygen as NADH with the concomitant production of oxidative stress.

**Figure 2 antioxidants-12-00673-f002:**
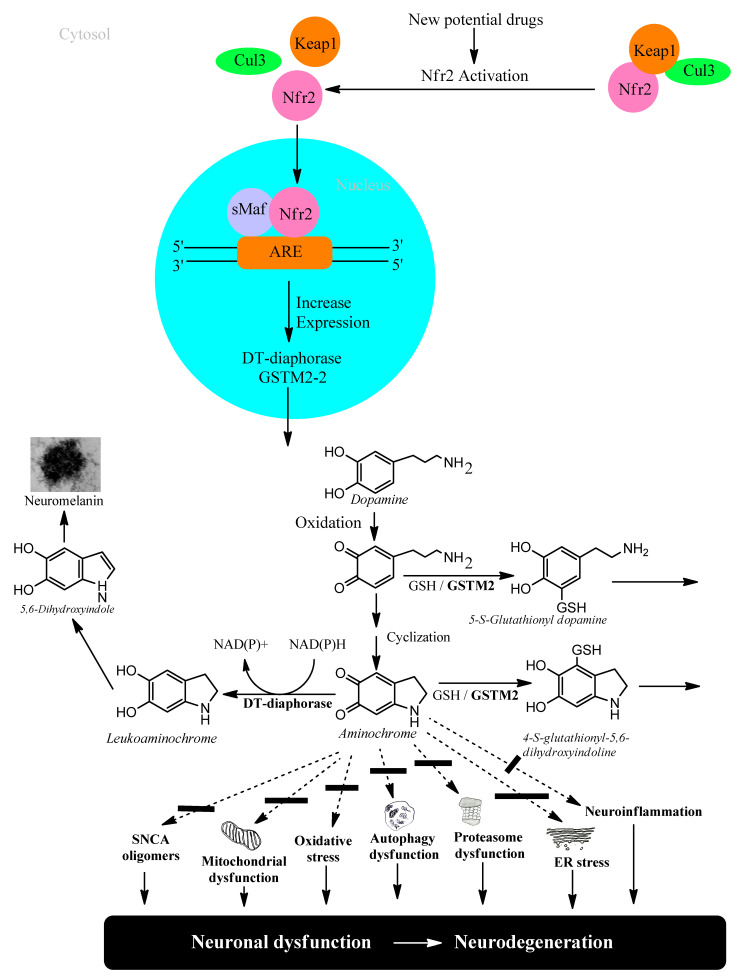
NFR2 activation to protect dopaminergic neurons from aminochrome neurotoxicity.

**Figure 3 antioxidants-12-00673-f003:**
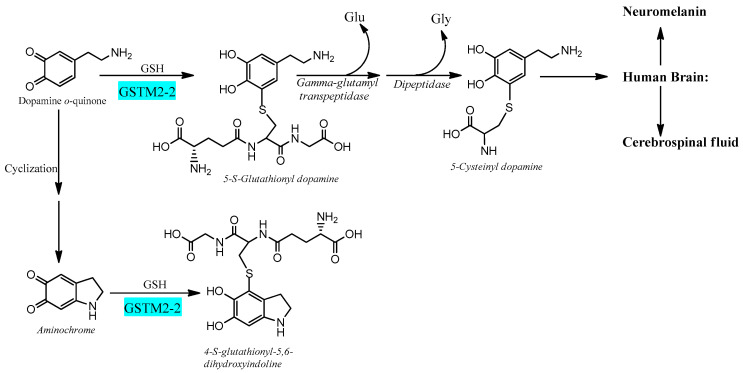
Glutathione transferase M2-2 catalyzes glutathione conjugation of ortho-quinones.

**Figure 4 antioxidants-12-00673-f004:**
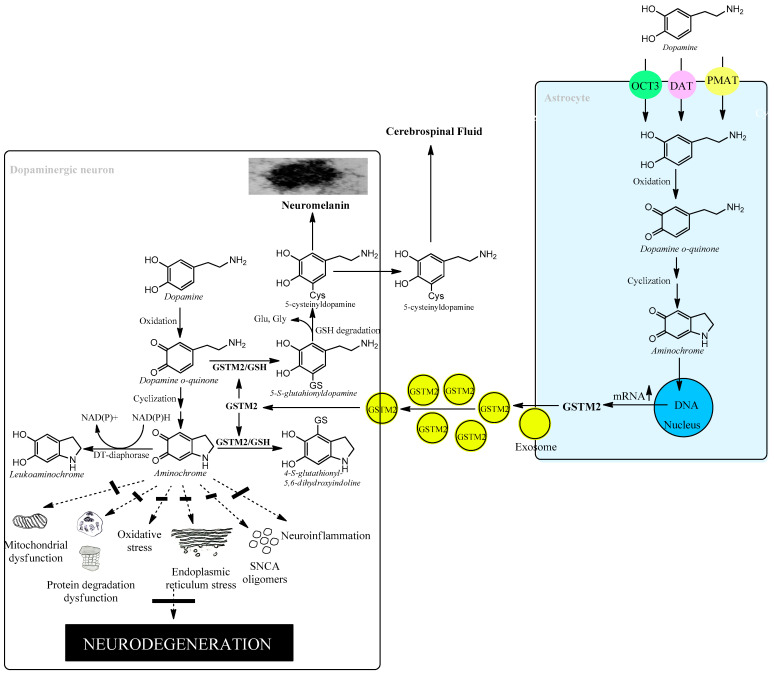
Astrocytes protect dopaminergic neurons against aminochrome neurotoxicity.

## Data Availability

Not applicable.

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
