# Peer review of "A Preclinical Model for Parkinson’s Disease Based on Transcriptional Gene Activation via KEAP1/NRF2 to Develop New Antioxidant Therapies"

_antioxidants, 2023, doi:10.3390/antiox12030673_

Round 1

Reviewer 1 Report

This manuscript reviews the difficulties that exist in the Parkinson's disease field to find new therapies that can inhibit or slow down the degenerative process of the nigrostriatal system in idiopathic Parkinson's disease. The authors propose that the degenerative process must be triggered by an endogenous neurotoxin and propose that the endogenous aminochrome neurotoxin that is generated during the synthesis of neuromelanin is responsible for inducing the degenerative process of the nigrostriatal system since it is formed inside the neurons that are lost during the disease. It does not have an expansive effect and is also able of induce all mechanisms related to this degenerative process. Furthermore, they propose that Nrf2 activation could be a way to increase the expression of these neuroprotective enzymes. This review introduces new ideas about Parkinson's disease that deserve to be published in order to open a discussion.

 However, the manuscript can be improved by revision according to these aspects.

1) Writing a more extensive paragraph about natural products that activate Nrf2 system.

2)  Adding a figure explaining the astrocytes protective role against aminochrome neurotoxicity.

3) It is mentioned many times neuromelanin. Then authors should briefly explain main features of neuromelanin structure and composition (cite Biesemeier A et al J Neurochem 2016; Engelen M et al PlosOne 2012).

4) L 101. Neuromelanin in neurons can be protective with its synthesis and toxic when released by dying neurons (quote these paper Zucca et al IUBMB Life 2023). 

5) Line 119. About toxines inducing PD like neurodegeneration cite also Capucciati et al Antioxidants 2021.

6) Line 266. Here mention that neuromelanin released by degenerating neurons in PD activate microglia and activated microglia induce degeneration of more neurons thus producing a vicious cycle of neuroinflammation and neurodegeneration (cite Zhang W et al  Neurotox Res 2011).

7) Line 336. In this model it is produced a melanin pigment like that of skin whose composition is very different from neuromelanin.

8) Line 355. Authors should mention that highly specific and sensitive mass spec studies showed that tyrosinase is not present in human Substantia Nigra ( Zucca et al npj Parkinson Disease 2018; Tribl F et al J Neural Transm Suppl. 2007).

9) Line 423. Here quote also Zucca FA et al npj Parkinson Disease 2018).

Author Response

Review 1

This manuscript reviews the difficulties that exist in the Parkinson's disease field to find new therapies that can inhibit or slow down the degenerative process of the nigrostriatal system in idiopathic Parkinson's disease. The authors propose that the degenerative process must be triggered by an endogenous neurotoxin and propose that the endogenous aminochrome neurotoxin that is generated during the synthesis of neuromelanin is responsible for inducing the degenerative process of the nigrostriatal system since it is formed inside the neurons that are lost during the disease. It does not have an expansive effect and is also able of induce all mechanisms related to this degenerative process. Furthermore, they propose that Nrf2 activation could be a way to increase the expression of these neuroprotective enzymes. This review introduces new ideas about Parkinson's disease that deserve to be published in order to open a discussion.

 However, the manuscript can be improved by revision according to these aspects.

  • Writing a more extensive paragraph about natural products that activate Nrf2 system.

Answer: Done as suggested

  • Adding a figure explaining the astrocytes protective role against aminochrome neurotoxicity.

Answer: Done as suggested

  • It is mentioned many times neuromelanin. Then authors should briefly explain main features of neuromelanin structure and composition (cite Biesemeier A et al J Neurochem 2016; Engelen M et al PlosOne 2012).

Answer; Done as suggested

  • L 101. Neuromelanin in neurons can be protective with its synthesis and toxic when released by dying neurons (quote these paper Zucca et al IUBMB Life 2023). 

Answer: Done as suggested

  • Line 119. About toxines inducing PD like neurodegeneration cite also Capucciati et al Antioxidants 2021.

Answer: Done as suggested

  • Line 266. Here mention that neuromelanin released by degenerating neurons in PD activate microglia and activated microglia induce degeneration of more neurons thus producing a vicious cycle of neuroinflammation and neurodegeneration (cite Zhang W et al  Neurotox Res 2011).

Answer: Done as suggested

  • Line 336. In this model it is produced a melanin pigment like that of skin whose composition is very different from neuromelanin.

Answer: Done as suggested

  • Line 355. Authors should mention that highly specific and sensitive mass spec studies showed that tyrosinase is not present in human Substantia Nigra ( Zucca et al npj Parkinson Disease 2018; Tribl F et al J Neural Transm Suppl. 2007).

Answer: Done as suggested

  • Line 423. Here quote also Zucca FA et al npj Parkinson Disease 2018).

Answer: Done as suggested

Reviewer 2 Report

In this article, Segura-Aguilar et al reviewed different preclinical PD model used for antioxidant therapies development, and a promising target Kcap1/NFR2 for protective therapeutic. This review provides many reasonable opinions on why the preclinical models used now are insufficient for develop antioxidant therapies and Kcap1/NFR2 activator as a promising antioxidant therapy target. Some revisions on structure, content and figure are needed. 

Structure:

1.     The abstract could be shortened and add an introduction section could provide more information about this article.

2.     Both the abstract and conclusion sections mention the model before the antioxidant therapy target, but the content is a bit disorganized. The mechanism of antioxidants and targeting pathway for PD treatment is discussed first, then the models, and then the mechanism of antioxidants again. It is better to adjust the order of sections and make the logic more fluent. Specifically, the “Mechanisms involved in neurodegeneration in Parkinson’s disease” should be placed first, followed by sections discussing ht mechanism of antioxidants, and  then transitioning to clinical models and potential target pathways for treatment. 

3.     The 6-Hydroxydopamine model is better to discuss in "Exogenous neurotoxins in preclinical models for Parkinson’s disease" section than in “Clinical studies to develop antioxidant therapies” section.

4.     The structure of the "Endogenous neurotoxins as preclinical models for Parkinson’ disease" section needs to be adjusted. It is better to first discuss the three lessons drawn, and then talk about each model specifically. Consider using subsections to better organize much of the background information, specifically in associating degeneration mechanisms to PD models.

5.     The relevance of the “Glutathione and oxidative stress” , “Glutathione transferases”, and “Transcriptional gene activation via KEAP1/NRF2” sections is not immediately evident based on the article title, until discussion of the amino chrome model. It might be beneficial to discuss this background following introduction of the aminochrome model. 

6.     The order of content in the article does not follow the logic of abstract, which confuses the primary line of reasoning. It is better to organize the topics of article to better reflect the reasoning established in the abstract. 

Content:

1.     Since two major parts are discussed in this article -- models and antioxidant targets -- a better title that includes both parts could be used. Current title seems contradictory as the authors present a model that is stated to be a good model of PD. It is better ot have a title that reflects the focus on (1) the aminochrome model as an accurate model of PD neurodegeneration, and (2) potential antioxidant targets arising from this model.

2.     In the section “Clinical studies to develop antioxidant therapies,” it seems that every failed clinical trial is based on the MPTP preclinical study. It is better to add some preclinical studies that used other models that also failed in the following clinical studies. This will help the audience see that all these models are not predictable for the treatments.

3.     More discussion about the alpha-synuclein model should be added, especially on why it is not suitable for antioxidant therapies development. Otherwise, it seems that the alpha-synuclein model cannot even be used as a PD model based on the current discussion.

4.     The abstract could be less detailed and shortened. It is better to discuss many of the mechanistic details currently in the abstract within the article itself. The title and abstract are incoherent and it is unclear what, specifically, the authors plan to discuss or assert after reading these.

5.     The relevance of the “Astrocytes protect dopaminergic neurons against aminochrome toxicity” to the larger point of the paper. It is better to discuss how this fact can be utilized in developing therapeutics or making this section a part of “Neuroprotection against neurodegeneration of nigrostriatal system in Parkinson’s Disease” rather than its own section. 

6.     There is a typo in the abstract, Line 35: "and Therefore" should be "Therefore."

7.     Several typos in the initial two sections. 

Figure:

1.     A graphical abstract could be added.

2.     Add more figures for each model discussed would help better demonstrate the advantages and disadvantages of those models as a preclinical model to develop antioxidant therapies. Alternatively, a table can be used to demonstrate the pros and cons in different models and in what studies they have been used.

3.     Figure 3 should be placed in the “Transcriptional gene activation via KEAP1/NRF2” section instead of the conclusion section.

4.     A figure could be added demonstrating the mechanism of degeneration in PD and at which point each model discussed attempts to mimic the disease (e,g, symptoms vs. pathology). 

Author Response

In this article, Segura-Aguilar et al reviewed different preclinical PD model used for antioxidant therapies development, and a promising target Kcap1/NFR2 for protective therapeutic. This review provides many reasonable opinions on why the preclinical models used now are insufficient for develop antioxidant therapies and Kcap1/NFR2 activator as a promising antioxidant therapy target. Some revisions on structure, content and figure are needed. 

Structure:

  1. The abstract could be shortened and add an introduction section could provide more information about this article.

Answer: Done as suggested

  1. Both the abstract and conclusion sections mention the model before the antioxidant therapy target, but the content is a bit disorganized. The mechanism of antioxidants and targeting pathway for PD treatment is discussed first, then the models, and then the mechanism of antioxidants again. It is better to adjust the order of sections and make the logic more fluent. Specifically, the “Mechanisms involved in neurodegeneration in Parkinson’s disease” should be placed first, followed by sections discussing ht mechanism of antioxidants, and  then transitioning to clinical models and potential target pathways for treatment. 

Answer: Done as suggested

  1. The 6-Hydroxydopamine model is better to discuss in "Exogenous neurotoxins in preclinical models for Parkinson’s disease" section than in “Clinical studies to develop antioxidant therapies” section.

Answer: Done as answer

  1. The structure of the "Endogenous neurotoxins as preclinical models for Parkinson’ disease" section needs to be adjusted. It is better to first discuss the three lessons drawn, and then talk about each model specifically. Consider using subsections to better organize much of the background information, specifically in associating degeneration mechanisms to PD models.

Answer: Done as suggested

  1. The relevance of the “Glutathione and oxidative stress” , “Glutathione transferases”, and “Transcriptional gene activation via KEAP1/NRF2” sections is not immediately evident based on the article title, until discussion of the amino chrome model. It might be beneficial to discuss this background following introduction of the aminochrome model. 

Answer: We have changed the title of this manuscript and done changes suggested

  1. The order of content in the article does not follow the logic of abstract, which confuses the primary line of reasoning. It is better to organize the topics of article to better reflect the reasoning established in the abstract. 

Answer: Done as suggested

Content:

  1. Since two major parts are discussed in this article -- models and antioxidant targets -- a better title that includes both parts could be used. Current title seems contradictory as the authors present a model that is stated to be a good model of PD. It is better ot have a title that reflects the focus on (1) the aminochrome model as an accurate model of PD neurodegeneration, and (2) potential antioxidant targets arising from this model.

Answer: Title: A preclinical model for Parkinson´s disease that can use that use transcriptional gene activation via KEAP1/NRF2 to develop new antioxidant therapies

  1. In the section “Clinical studies to develop antioxidant therapies,” it seems that every failed clinical trial is based on the MPTP preclinical study. It is better to add some preclinical studies that used other models that also failed in the following clinical studies. This will help the audience see that all these models are not predictable for the treatments.

Answer: Done

  1. More discussion about the alpha-synuclein model should be added, especially on why it is not suitable for antioxidant therapies development. Otherwise, it seems that the alpha-synuclein model cannot even be used as a PD model based on the current discussion.

Answer: To test a new drug that can change the course of the disease or inhibit the progress of the degenerative process, it is required that the preclinical model include the neurotoxin that triggers all the mechanisms involved in the loss of neuromelanin-containing dopaminergic neurons. Hence the failure of clinical studies using antioxidants, because what induces oxidative stress in patients with idiopathic Parkinson's was not included in these preclinical model. Alpha-synuclein is neurotoxic by itself in its monomeric form but when neurotoxic oligomers are formed. In the case of familial Parkinson's, it is a mutation that triggers the formation of neurotoxic oligomers, but in idiopathic Parkinson's, the only thing known to date triggers the formation of neurotoxic alpha-synuclein oligomers is aminochrome. For this reason, preclinical models with alpha-synuclein mutations are not suitable for testing new drugs or antioxidants for patients with idiopathic Parkinson's.

  1. The abstract could be less detailed and shortened. It is better to discuss many of the mechanistic details currently in the abstract within the article itself. The title and abstract are incoherent and it is unclear what, specifically, the authors plan to discuss or assert after reading these.

Answer: we have changed the title to “Aminochrome as preclinical model for Parkinson´s disease that can use transcriptional gene activation via KEAP1/NRF2 to develop new antioxidant therapies”

  1. The relevance of the “Astrocytes protect dopaminergic neurons against aminochrome toxicity” to the larger point of the paper. It is better to discuss how this fact can be utilized in developing therapeutics or making this section a part of “Neuroprotection against neurodegeneration of nigrostriatal system in Parkinson’s Disease” rather than its own section. 

Answer: Done as suggested

  1. There is a typo in the abstract, Line 35: "and Therefore" should be "Therefore."

Answer: removed

  1. Several typos in the initial two sections. 

Answer: Done

Figure:

  1. A graphical abstract could be added.

Answer: done

  1. Add more figures for each model discussed would help better demonstrate the advantages and disadvantages of those models as a preclinical model to develop antioxidant therapies. Alternatively, a table can be used to demonstrate the pros and cons in different models and in what studies they have been used.

Answer:

  1. Figure 3 should be placed in the “Transcriptional gene activation via KEAP1/NRF2” section instead of the conclusion section.

Answwer: done

  1. A figure could be added demonstrating the mechanism of degeneration in PD and at which point each model discussed attempts to mimic the disease (e,g, symptoms vs. pathology). 

Answer: We have done one more figure.

Round 2

Reviewer 2 Report

The authors have addressed the comments accordingly, and I would suggest the acceptance.